# Genomic and Transcriptomic Approaches Provide a Predictive Framework for Sesquiterpenes Biosynthesis in *Desarmillaria tabescens* CPCC 401429

**DOI:** 10.3390/jof9040481

**Published:** 2023-04-17

**Authors:** Tao Zhang, Jianjv Feng, Wenni He, Xiaoting Rong, Hui Lv, Jun Li, Xinxin Li, Hao Wang, Lu Wang, Lixin Zhang, Liyan Yu

**Affiliations:** 1Institute of Medicinal Biotechnology, Chinese Academy of Medical Sciences & Peking Union Medical College, Beijing 100050, China15524221591@163.com (H.L.);; 2State Key Laboratory of Bioreactor Engineering, East China University of Science & Technology, Shanghai 200237, China

**Keywords:** *Desarmillaria*, genome mining, phylogeny, sesquiterpene synthases, RNA sequencing, functional identification

## Abstract

Terpenoids constitute a structurally diverse class of secondary metabolites with wide applications in the pharmaceutical, fragrance and flavor industries. *Desarmillaria tabescens* CPCC 401429 is a basidiomycetous mushroom that could produce anti-tumor melleolides. To date, no studies have been conducted to thoroughly investigate the sesquiterpenes biosynthetic potential in *Desarmillaria* or related genus. This study aims to unravel the phylogeny, terpenome, and functional characterization of unique sesquiterpene biosynthetic genes of the strain CPCC 401429. Herein, we report the genome of the fungus containing 15,145 protein-encoding genes. MLST-based phylogeny and comparative genomic analyses shed light on the precise reclassification of *D*. *tabescens* suggesting that it belongs to the genus *Desarmillaria*. Gene ontology enrichment and pathway analyses uncover the hidden capacity for producing polyketides and terpenoids. Genome mining directed predictive framework reveals a diverse network of sesquiterpene synthases (STSs). Among twelve putative STSs encoded in the genome, six ones are belonging to the novel minor group: diverse Clade IV. In addition, RNA-sequencing based transcriptomic profiling revealed differentially expressed genes (DEGs) of the fungus CPCC 401429 in three different fermentation conditions, that of which enable us to identify noteworthy genes exemplified as STSs coding genes. Among the ten sesquiterpene biosynthetic DEGs, two genes including *DtSTS9* and *DtSTS10* were selected for functional characterization. Yeast cells expressing DtSTS9 and DtSTS10 could produce diverse sesquiterpene compounds, reinforced that STSs in the group Clade IV might be highly promiscuous producers. This highlights the potential of *Desarmillaria* in generating novel terpenoids. To summarize, our analyses will facilitate our understanding of phylogeny, STSs diversity and functional significance of *Desarmillaria* species. These results will encourage the scientific community for further research on uncharacterized STSs of Basidiomycota phylum, biological functions, and potential application of this vast source of secondary metabolites.

## 1. Introduction

Sesquiterpenes are synthesized from the prenyl precursors farnesyl diphophate (FPP) by a family of enzymes known as sesquiterpene synthases (STSs) [1]. Basidiomycete mushrooms are particularly adept at producing a wide range of structurally complexified sesquiterpenes [2]. Exploration of the medicinal and pharmacological potential of these underexplored mushroom species could be scientifically and pharmacologically attractive. Continuing secondary metabolites discovery targeting rare basidiomycetes, previous investigations have reported a number of active terpenoids or meroterpenoids, such as antroalbocin A [3], melleolides [4], illudin S [5], conosilane A [6], sterhirsutins [7], many of which exhibit some antibiotic and cytotoxic properties. The ever-increasing genomic information has revealed great potential for mining of putative STS homologues, indicating that the phylum Basidiomycota terpenoids and their synthases represent abundant but largely untapped natural resources [8,9]. Specially, each fungus of the phylum Basidiomycota has an average of 10–20 putative STSs encoding genes. However, only few sesquiterpenes are reportedly characterized [4,7]. Intriguingly, the basidiomycetous STSs have been shown to be readily expressed in well-established laboratory hosts, i.e., *Escherichia coli* [8,10] and *Saccharomyces cerevisiae* [11]. Moreover, as many basidiomycetes are relatively slow growing in laboratory conditions. Thus, all this makes heterologous expression an attractive approach for accessing the basidiomycete terpenoid chemistry.

Basidiomycetous *Armillaria*, belonging to the family Physalacriaceae family, are ubiquitous as destructive forest pathogens of forests, orchards, and plantations [12]. Species from this genus cause the *Armillaria* butt and root disease that weakens and often kills woody perennials [12,13]. *Armillaria* species were considered to be polymorphic, which impact on forest populations has both ecological and economic significance [13]. There are approximately more than 70 officially described species [14], and accurate species delimitation and pathogenicity levels assessment of *Armillaria* genus is pivotal for forestry conservation [15]. As saprotrophs, *Armillaria* species produce fleshy fruiting bodies that appear in large clumps around contaminated plants [16]. The fungus is beneficial in some ways, as it serves as a natural recycler of nutrients in wooded ecosystems, breaking down woody materials to increase soil health. Interestingly, *Armillaria* species are also considered as source of nutrition in Europe and Asia and some species are associated with traditional Chinese medicine for their pharmaceutical properties [17]. *Armillaria* spp. were demonstrated to yield a wealth of biologically active metabolites, exemplified as sterols, sesquiterpene aryl esters, and many other components exhibiting therapeutic or dietary values [17,18,19]. Thus, it is obligated to fully elucidate the molecular basis or mechanism of *Armillaria* at the genomic level.

The technological revolution in genome sequencing and bioinformatics have markedly increased the number of sequenced Basidiomycetous genomes. There are already nine *Armillaria* genomes that have been published to date, and those for *A*. *mellea* DSM3731, *A*. *ostoyae* C18/9, *A*. *solidipes* 28-4, *A*. *borealis* FPL87.14 v1.0, *A*. *fuscipes* CMW2740, *A*. *cepistipes* B5, *A*. *altimontana* 837-10, two *A*. *gallica* strains Ar21-2 and 012m were included. Analyses of these genomic and transcriptomic information have provided insights into evolution, genetics, developmental biology, plant-cell-wall-degrading enzymes (PCWDE) repertoire, and related pathogenicity factors [20,21,22,23,24]. Sipos and coworkers performed omics sequencing to provide genetic and regulatory insights into toolkits for wood-decaying, morphogenesis and complex multicellularity including rhizomorph and fruiting bodies development of the strain *A*. *ostoyae* C18 [23]. The taxonomy of *Armillaria* has raised controversial debate, recently, *Armillaria* was narrowed down to include only annulate species, and *Desarmillaria* was introduced to accommodate two exannulate mushroom-forming armillarioid species (*D*. *ectypa* and *D*. *tabescens*) [25,26]. The lineage to diverge is that composed of annulate mushroom-forming armillarioid species, in what *A*. *mellea* (Vahl: Fr.) P. Kumm is the type species [27]. Koch et al. (2017) conducted six-locus phylogenetic analysis among *Armillaria* and related affinities, demonstrated that *Guyanagaster* and *Desarmillaria* are close lineages [25]. On the delimitation and characterization of *Armillaria* species, the genome-based phylogenomics and comparative genomics were invaluable to understand the phylogeny of *Armillaria* and sister groups.

In this study, we report a 50.36 Mb draft genome sequence of *D. tabescens* CPCC 401429. MLST-based phylogeny and synteny analysis at the genomic level were investigated, which reinforced that the *D*. *tabescens* was supposed to belong to the genus of *Desarmillaria* instead of *Armillaria* as it used to. Genome profiling and transcript on three different fermentation conditions shed light on the secondary metabolism. Notably, guided by STSs phylogeny based predictive framework, a diverse network of twelve putative sesquiterpene biosynthetic genes were obtained and half of them were categorized into the new subfamily, i.e., diverse Clade IV. Intriguingly, two genes including *DtSTS9* and *DtSTS10* were codon optimized and cloned for heterologous expression in yeast host. Through gas chromatography-mass spectrometry (GC-MS) analysis, yeast cells expressing DtSTS9 and DtSTS10 produce diverse sesquiterpenes. This work provides the genomic basis to understand and investigate the new genus *Desarmillaria*. We exemplify the exploitation on uncharacterized STSs of basidiomycetes and help to broaden our knowledge of sesquiterpenes biosynthesis.

## 2. Materials and Methods

### 2.1. Strains and Cultivation Conditions

The fungus *D. tabescens* CPCC 401429 was cultured on potato dextrose medium (PDA; BD company) and incubated at 28 °C for two weeks [4]. To produce mycelium for genomic DNA (gDNA) extraction, the strain was grown in liquid MEP medium (malt extract broth 2.0%, BD, soy bean powder 0.2%). For RNA sequencing, the strain CPCC 401429 was cultivated in PDB broth (BD), YMEG broth (malt extract 1%, yeast extract 0.4%, glucose 0.4%, CaCO_3_ 0.2%), and F1 broth (corn steep liquor 1.0%, soy bean powder 0.2%, glucose 2.0%, peptone 0.5%, ammonium sulfate 0.2%), respectively. The cultures were cultivated on a rotatory shaker (130 rpm) for 15-day.

Standard DNA cloning experiments were conducted using *E*. *coli* Trans1-T1. *E*. *coli* cultures carrying respective plasmid were grown in Luria-Bertani (LB) medium supplemented with ampicillin (100 μg/mL). *S*. *cerevisiae* BJ5464 was used as the host for heterologous expression [28]. Individual transformant of the yeast strain was grown in shaking cultures in 10 mL synthetic dextrose minimal medium containing 6.7 g/L of yeast nitrogen base without amino acids, 20 g/L of dextrose, 1.3 g/L of amino acid dropout powder. For production of the sesquiterpenes, a total of 200 mL YPD broth (BD) was inoculated with 5 mL of the starter culture and incubated for 96 h at 30 °C under shaking conditions (200 rpm). The strains, plasmids, primers, and genetically engineered cells are listed in Table 1.

### 2.2. Sequencing, De Novo Assembly, and Bioinformatic Tools

Genomic DNA from *Desarmillaria* was isolated using HP Fungal DNA Kit (Omega, Guangzhou, China) according to the manufacturer’s instructions. Further purification was conducted using RNase digestion, chloroform extraction, and isopropanol precipitation [29]. The high molecular weight gDNA was washed twice with 75% ethanol, and dissolved in 200 μL RNase-free water. Integrity and amount of the extracted gDNA were detected by 1% agarose gel electrophoresis and using the NanoDrop ND-2000 Spectrophotometer (Thermo Fisher, Waltham, MA, USA). Shotgun sequencing was performed in the laboratory of at Shanghai Majorbio Bio-pharm Technology Co. Ltd. (Shanghai, China) using Illumina Hiseq 2000 platform. The PE150 (pair-end) libraries with a mean insert size 400 bp were subjected to PacBio sequencing. Adapter sequences were trimmed and short reads were formatted using Trimmomatic v. 0.36 [30] with minimum quality of 19 and minimum length of 30 bp. Quality was assessed using SMRT Analysis software [31]. 

The obtained sequence reads were assembled into contigs and scaffolds using SOAPdenovo v. 2.04 [32] and Canu v. 1.7 [33]. K-mer values were automatically selected based on read length and data type. The final scaffold set was verified for miss-assemblies using the BUSCO v3.0 and corrected if necessary [34]. Gene annotations of the coding sequences were formatted to GenBank database for BLASTP alignment or obtained via AUGUSTUS program [35] and verified on the basis of homologous proteins in the NCBI database. The gene ontology (GO) functional annotation was performed using Blast2go [36]. In parallel, protein annotations were also performed using KEGG (Kyoto Encyclopedia of Genes and Genomes), Pfam [37], and Swiss-prot databases [38]. 

### 2.3. RNA Purification, Library Construction, and RNA-Sequencing

The total RNA was extracted from PDB broth, YMEG broth, and F1 broth using PureLink^TM^ RNA Mini Kit (Invitrogen, USA) and gDNA was removed using DNaseI (Takara). The RNA quality and concentration measurements were performed using 2100 Bioanalyzer (Agilent) and quantified using the NanoDrop ND-2000, and only high-quality purified RNA was processed for construction sequencing library (OD_260/280_ = 1.8–2.2, OD_260/230_ ≥ 2.0, RIN ≥ 8.0, 28S:18S ≥ 1.0, >1 μg). RNA purification, reverse transcription, library construction and sequencing were performed at Shanghai Majorbio Bio-pharm Biotechnology Co., Ltd. (Shanghai, China) based on the manufacturer’s instructions (Illumina). Double-stranded cDNA libraries construction were carried out using the TruSeq^TM^ RNA sample preparation Kit using 1.0 μg of total RNA (Illumina). Libraries were size selected for cDNA target fragments of 300 bp on 2% Low Range Ultra Agarose followed by PCR amplified using Phusion DNA polymerase (NEB) for 15 PCR cycles. Sequencing was conducted on Illumina NovaSeq 6000 sequencer instrument (2 × 150 bp read length). 

### 2.4. Quality Control, Read Mapping, and Transcriptome Analysis 

The raw paired end reads were trimmed and quality controlled by FASTQ preprocessor with default parameters [39]. Then clean reads were separately aligned to reference genome with orientation mode using HISAT2 software [40]. The mapped reads of each sample were assembled by StringTie in a reference-based strategy [41]. To identify differential expression genes (DEGs) between two different samples, the expression level of each gene was calculated according to the transcripts per million reads (TPM) approach. RSEM was used to quantify gene abundances [42]. Functional-enrichment analysis including GO, KEGG were performed to identify which DEGs were significantly enriched in GO terms and metabolic pathways at P-adjust ≤ 0.05 compared with the whole-transcriptome background. GO functional enrichment and KEGG pathway analysis were performed by Goatools and KOBAS [43,44]. 

### 2.5. Multi-Locus Sequence Typing (MLST)-Based Phylogenetic Analysis 

To determine the phylogeny and identity of the strain *D. tabescens* CPCC 401429, we compiled a dataset composed of sequence data from twenty-five sequenced basidiomycetes including eight *Armillaria* spp. and sixteen Agaricales encompassing white and brown rot wood decomposers and ectomycorrhizal fungi, species of *Heterobasidion irregulare* TC 32-1 from Russulales served as outgoup [23]. Eleven loci including 18S-ITS-28S region, elongation factor 1-α (EF1α), RNA polymerase II (RPB2), actin-1 (ACT1), glyceraldehyde-3-phosphate dehydrogenase (GPD), beta-tubulin (TUB), plasma membrane H^+^-ATPase (ATP), glucose-6-phosphate isomerase (PGI), 60S ribosomal protein L2 (RPL), 60S ribosomal protein L18 (RPK), and polyubiquitin (UBQ) coding genes were performed, to delimit *Armillaria* systematic relationship through phylogenetic analysis [25,45]. The sequences used are compiled in Appendix A.

Sequences alignment were conducted using MAFFT v7.471 with FFT-NS-2 algorithm [46], followed by MEGA 7 to refine the alignment done manually [47]. The introns of the housekeeping genes were excluded. ModelFinder was used to predict the best-fitted substitution model according to Akaike Information Criterion (AIC) [48]. The model VT was selected for Bayesian inference (BI) analyses. Bayesian trees were constructed with MrBayes v3.2.6 under the GTR+G+I model [49]. Two independent runs were conducted for 10,000,000 generations with sampling every 100 generations using Markov Chain Monte Carlo (MCMC) algorithm. The first 25% of the samples were discarded as burn-in fraction. The Bayesian tree generated was visualized in FigTree v1.4.4 (http://tree.bio.ed.ac.uk/software/figtree/, accessed on 25 November 2018) and edited using Adobe Illustrator software.

### 2.6. Bioinformatic Annotation for STSs and Phylogenetic Tree Construction

The genome of *D*. *tabescens* CPCC 401429 was further bioinformatically analyzed using antiSMASH database. All the predicted amino acid sequences of protein-coding genes present in the genome of the strain CPCC 401429 have been searched for homologues by BLASTP tool in NCBI database. The predicted STSs genes used for the functional analysis were annotated and verified for a coherent exon/intron structure. In addition, several characterized fungal STSs from *Clitopilus pseudopinsitus*, *Omphalotus olearius*, *Agrocybe aegerita*, *Coniophora puteana*, *Coprinopsis cinerea*, *Postia placenta*, *Lignosus rhinocerotis*, *Stereum hirsutum*, *Termitoymces* sp., *Armillaria gallica*, *Fomitopsis pinicola*, and *Heterobasidion annosum* were incorporated for phylogenetic analyses. Bayesian trees of STSs were constructed with MrBayes v3.2.6 using the same parameters during MLST-based phylogenetic analysis [49].

### 2.7. Gene Synthesis, Cloning and Expression of STSs in Yeast

Candidate STSs coding genes from the strain CPCC 401429 were synthesized by Shanghai Generay Biotech Co., Ltd. and codon-optimized for heterologous expression in host *S. cerevisiae*. The oligonucleotide primers used for constructing the genes are listed in Table 1. PCR reactions were performed using Q5 High-Fidelity DNA Polymerase (New England Biolabs) according to the manufacturer’s protocol at an annealing temperature of 56 °C. In vivo yeast recombination cloning strategy was performed using a Frozen-EZ Yeast Transformation II Kit™ (Zymo Research) where competent cell *S. cerevisiae* BJ5464 was transformed with the individual DNA fragment (PCR product) of the respective STS gene and linearized vector backbone derived from YET plasmid [28,29]. Yeast transformants grown on synthetic dropout agar plate lacking tryptophan were screened by diagnostic PCR. Plasmid DNA was isolated from positive transformants using a E.Z.N.A^®^ Yeast Plasmid Miniprep Kit (Omega, Guangzhou, China) and used to transform chemically competent *E. coli* Trans1-T1 (TransGen Biotech, Beijing, China). Plasmid DNA was isolated from the positive clones grown on LB broth with ampicillin using E.Z.N.A^®^ Plasmid DNA Mini Kit (Omega, Guangzhou, China) and further verified by DNA sequencing.

### 2.8. GC-MS Analysis of Terpenoids

After fermentation, the *S. cerevisiae* liquid cultures (5.0 g each) were harvested in the headspace (20 mL) and the samples were placed at 55 °C for 15 min and volatile compounds were sampled at 55 °C for 40 min by SPME with a DVB/CAR/PDMS fiber. Compounds were desorbed in the split/splitless inlet of an Agilent 7890B/5977A gas chromatography equipped with an Agilent 7200 accurate-mass quadrupole time-of-flight (GC-MS-TOF; Agilent Technologies, Santa Clara, CA, USA) and analyzed at 250 °C for 5 min. The GC/MS-TOF was equipped with a DB-WAX column (Agilent Technologies; 60 m × 0.25 mm ID, 0.25 μm film thickness) [8,11,12]. The system was operated under the following condition: compounds were detected in split mode at split ratio of 0:1; the GC oven temperature was programmed to ramp from 50 °C (held for 5 min) to 90 °C at 3 °C min^−1^, to 150 °C at 2 °C min^−1^, then to 230 °C at 8 °C min^−1^, and finally to 230 °C (held for 5 min). Full scan mass spectra were acquired in the mass range of 50–350 *m/z*, and ionization was performed by electron impact at 70 eV with an electrospray ionization source temperature set at 230 °C.

## 3. Results

### 3.1. General Genome Features of D. tabescens CPCC 401429

With the Illumina PE150-based HiSeq 2500 sequencer and PacBio Sequel, raw data containing 871,317 reads and 6147.6 million bases were obtained. The high-quality reads were assembled by SOAPdenovo 2.04 and Canu 1.7 to generate 713 polished scaffolds. According to the coverage and the GC content, the assembly were performed and optimized to afford the draft genome sequence of *D*. *tabescens* CPCC 401429. The genome was 50.36 Mb in size and included 705 scaffolds with an N_50_ of 126 kb and a GC content of 47.14% (Table 2). There were approximately 0.06% (29,142 bp) repetitive sequences in the genome. A total of 15,415 protein-coding genes were predicted in the genome by AUGUSTUS program [35]. The genome completeness was 92.4% based on fungal lineage employed by the BUSCO 3.0 assessment [50]. In addition, 255 tRNAs and 4 rRNAs were defined with tRNA-scan-SE 2.0 and Barrnap V 0.8, respectively. The genome size and number of predicted genes are markedly different with eight other *Armillaria* mushrooms (Appendix A). All the assembly data indicated that whole genome supports a detailed investigation of the gene content and sesquiterpene synthases-encoding pathway of sequenced *Armillaria* strains. Among the sequenced strains, the *D*. *tabescens* is the smallest, while others are >55 Mb, and even the strain of *A*. *gallica* 012m with 87.31 Mb. Significantly, a number of STSs encoding genes were detected among the genus, in which the strain CPCC 401429 represented the median level with twelve genes (Appendix A). 

### 3.2. Phylogenetic Analysis of the Strain CPCC 401429 

The *tef*-1*α* DNA sequence retrieved from the genome of CPCC 401429 strain was used to preliminarily determine the identity in a BLAST nucleotide search of NCBI and exactly matched the *tef*-1*α* region of *D*. *tabescens* HKAS86603 (GenBank accession KT822441) with 99.6% similarity. A phylogenetic dendrogram of *tef*-1*α* recovered from CPCC 401429 and other *Armillaria* species or lineages (Figure 1) was generated, as shown in this phylogenetic tree, the *D*. *tabescens* is more closely clustered to *D*. *ectypa* species and formed a distinct group (clade V). Interestingly, Koch et al. (2017) performed six loci (*28S*, *EF1α*, *RPB2*, *TUB*, *actin-1* and *gpd*) phylogenetic analysis to investigate and resolve the evolutionary status of diverging lineages *Guyanagaster* and *Armillaria* subgenus *Desarmillaria* [25]. To further investigate the armillarioid phylogeny, 18S-ITS-28S regions and ten housekeeping genes (*tef*-*1α*, *rpb2*, *act1*, *gpd*, *tub*, *atp*, *pgi*, *rpl*, *rpk* and *ubq*) were selected for MLST-based phylogenetic tree construction. As shown in Figure 2, markedly, the species *D*. *tabescens* CPCC 401429 and the species *Guyanagaster necrorhizus* MCA 3950 clustered as a well-supported branch. However, *Armillaria* sensu stricto includes annulate mushroom-forming armillarioid species, exemplified as *A*. *mellea*, *A*. *ostoyae*, *A*. *borealis*, *A*. *cepistipes*, *A*. *solidipes*, *A*. *fuscipes*, *A*. *altimontana*, and *A*. *gallica* species. Our result reinforced that *D*. *tabescens* was supposed to belong to the genus of *Desarmillaria* [25,51]. 

### 3.3. Genome Annotation and Bioinformatic Analysis

The data of genomic annotations were summarized in Appendix A. During the analysis of GO classification, 9044 genes were annotated with the three main categories: cellular component, molecular function, and biological process, respectively (Figure 3A). The identified coding proteins associated with molecular function are more than other two groups. In the KEGG pathway annotation, the 8300 predicated genes could be divided into 46 categories based on their functions (Figure 3B). Among these categories, 94 genes are associated with the metabolism of polyketides and terpenoids, which indicates that *D*. *tabescens* was a potential natural resource for polyketides and terpenoids biosynthesis. In the carbohydrate-active enzymes (CAZymes) annotation, 509 candidate CAZymes coding genes were identified in the genome of the strain CPCC 401429. Significantly, 199 glycoside hydrolases (GH) dominated in the classes of CAZymes, followed by 145 auxiliary activities proteins (AA), carbohydrate esterases (CE), glycosyl transferases (GT), polysaccharide lyases (PL), and carbohydrate-binding modules (CBM). Compared with the genome of other close *Armillaria* species, the strain CPCC 401429 had less CAZymes in which the GHs, GTs, and CBMs represented the smallest sets among six *Armillaria* strains (Appendix A). In addition, as saprotrophic wood-rotting fungi, *Armillaria* species exhibit higher quantities of CEs and AAs than other Agaricales exemplified as *Agaricus bisporus*, *Pleurotus ostreatus*, *Coprinopsis cinerea*, *Schizophyllum commune*, *Lentinula edodes*, and *Laccaria bicolor*. Interestingly, most CEs and AAs could be functioning as lignin-, cellulose-, pectin- and hemicellulose-degrading enzymes, indicating the strong potential to degrade plant cell wall (PCW) components. Cytochrome P450s, as multifunctional oxygenases, play a key role in adaptation to nutritional niche or secondary metabolites biosynthesis. Based on domain and pfam prediction, a total of 404 cytochrome P450s coding genes were screened in the genome. The number exceeded those of *A*. *gallica* 012m (*n* = 271), *A*. *cepistipes* B5 (*n* = 324) and *A*. *gallica* Ar21-2 (*n* = 330). These P450s might be closely related to the formation of basic metabolism and secondary metabolites biosynthesis in *D*. *tabescens* [4]. There are three major bioactive compounds in *D*. *tabescens*, exemplified as melleolides, armillamide, and ergosterol [4]. The synthesis of orsellinate-derived melleolides and armillamide are catalyzed by a cascade of redox reactions through the combined action of cytochrome P450s and oxidoreductases, respectively [4]. 

### 3.4. The Composition of STSs Homologues and Phylogeny of STSs 

Bioinformatic analysis using antiSMASH database and a BLASTP search for putative sesquiterpene synthases (STSs) present in the genome of the strain *D*. *tabescens* CPCC 401429 revealed twelve candidate sesquiterpene biosynthetic genes (*DtSTS1*–*DtSTS12*, accession no. OQ442799-OQ442810) coding for the STSs (Figure 4). Investigation of the flanking regions of the STSs gene allowed the discovery of other genes coding cytochrome P450 monoxygenase or SDR reductase. As shown in Figure 5, In the phylogenetic analysis, the STSs are clustered in three categories: (i) *Δ*6 -protoilludene type (Clade III), this group includes DtSTS3, DtSTS6, and DtSTS7, which cluster with Pro1 from *A*. *gallica* [52], PRO1_ARMOS from *A*. *ostoyae*, Arg6 and Arg7 from *Agrocybe aegerita* [10]. This specificity also counts for protoilludene synthases from *Omphalotus olearius* and *S*. *hirsutum*. It has been demonstrated that *Δ*6-protoilludene is the precursor for melleolides that have shown antitumor and antimicrobial activities [4]. (ii) DtSTS5, DtSTS11, and DtSTS12 are closely related to Agr1 and Agr3 from *A*. *aegerita* [10], Cop1-Cop3 from *C*. *cinereus* [8], and Omp3 from *Omphalotus olearius* [9]. These STSs have been characterized in previous genome-wide studies, in which *S*. *cerevisiae* and/or *E*. *coli* served as heterologous systems [11]. The STSs of this group could utilize a 1,10-cyclization of (2*E*, 6*E*)-farnesyl pyrophosphate (FPP) to generate sesquiterpenes derived from an (*E*, *E*)-germacradienyl cation. Cop1 and Cop2 synthesize germacrene A as the major product, while Cop3 and Omp1 are α-muurolene synthases [8,9]. Omp3 accumulated several sesquiterpenes including α-muurolene, β-elemene, δ-cadinene, and selina-4, 7-diene [8,9]. (iii) The residual six STSs were grouped into Clade IV, this group includes few members, i.e., Cop6 from *C*. *cinereus*, ShSTS1 from *S*. *hisutum*, Omp9 and Omp10 from *O*. *olearius*. Cop6 is *α*-cuprenene synthase [8], Omp9 could synthesize α-barbatene (57%) and β-barbatene (21%), while Omp10 synthesizes daucene (21%), trans-dauca-4 (11),8-diene (71%) [9]. This implied that STSs in Clade IV might produce diverse sesquiterpene compounds.

### 3.5. RNA-Sequencing Based Transcriptome Analysis of the Strain CPCC 401429

To further investigate and evaluate the differently expressed genes (DEGs), we constructed three DEGs libraries to compare YMEG to F1, PDB to YMEG, and PDB to F1 (Appendix A). Overall, there are 10,003, 10,623, and 9895 genes which were transcribed in F1, PDB, and YMEG fermentation broth, respectively. Among the transcribed genes, 9367 genes were commonly shared (Figure 6A). Additionally, we detected 1721, 1531, and 1616 up-regulated genes, and 1948, 2271, and 2026 down-regulated genes between PDB and YMEG libraries, PDB and YMEG libraries, PDB and F1 libraries, respectively (Figure 6B, Appendix A). In organisms, different genes accomplish in a synchronizing mode to perform the biological functions. Pathway-directed analysis supports to further characterize the molecular functions of genes. Functional enrichment analysis was conducted using all DEGs against the GO database to investigate DEGs involved in metabolism and development. The enrichment data supports the hypothesis that the DEGs are approximately related to glycoside metabolism, fruiting body formation, membrane transporters, and secondary metabolites (SMs) production (Appendix A). The latter includes SMs biosynthesis, exemplified as meroterpenoid melleolides, polyketides, and sesquiterpenes. It is worth noting that the DEGs functioning in the basal metabolic process, e.g., glycoside metabolism and transmembrane transport, which could promote the biological process of secondary metabolism (Appendix A). Moreover, KEGG pathway analysis indicated that mevalonate and acetyl-CoA and metabolic pathways were also up-regulated, which facilitate the biosynthesis of sesquiterpenes and melleolides. Ten genes associated with GO terms related to sesquiterpene biosynthesis were identified. Among them, one gene *DtSTS3* exhibited up-regulation in F1 fermentation broth, which is responsible for melleolides biosynthesis. While five genes (*DtSTS4*, *DtSTS7*, *DtSTS9*, *DtSTS11*, *DtSTS12*) and four genes (*DtSTS2*, *DtSTS5*, *DtSTS8*, and *DtSTS10*) showed up-regulated expression in PDB broth and YMEG broth, respectively (Figure 6C). We envisage that these transcriptome analysis results would be conductive to the further study of STSs characterization in *Desarmillaria* or close affinities.

### 3.6. Heterologous Expression of STSs Encoding Genes in Yeast Produced Diverse Sesquiterpenes 

To elucidate the sesquiterpenes (STs) potentially produced by the strain CPCC 401429, we first attempted to reconstitute STs production using the *S*. *cerevisiae* BJ5464 strain as a heterologous host. Two predicted STSs (DtSTS9, DtSTS10) were codon optimized, cloned into the YET vector and verified by sequencing. The expression plasmids were transformed into the BJ5464 strain [28]. For this, 100 mL of cultures were grown to mid log phase and then solid phase micro-extraction (SPME) coupled with Gas Chromatography Mass Spectrometry (GC-MS) was used to extract and analyze the volatiles in the headspace. Subsequent analyses of terpenoids produced by the resulting recombinant transformants carrying corresponding STSs expression cassette were performed. The two DtSTSs gave rise to diverse sesquiterpenes in liquid cultures of the corresponding yeast transformant (Table 3), while no sesquiterpenes was detected from the control strain BJ-CK. Terpene compounds were identified by comparing retention indices and mass spectra to National Institute of Standards and Technology (NIST) Standard Reference Database (v20, 2023-01, https://webbook.nist.gov/, accessed on 25 November 2018). Comparison of the GC–MS profiles of the yeast cultures expressing DtSTS9 and DtSTS10 against the empty vector yeast control revealed the putative sesquiterpene (*m/z* = 204, C_15_H_24_), or sesquiterpene alcohols (*m/z* = 240, C_15_H_26_O) as products (Table 3). Wide varieties of sesquierpenes were detected in the yeast culture expressing DtSTS9 and DtSTS10. Besides the noticeable product cis-thujopsene (11.25% of the peak area), β-himachalene (4.82%) as well as small amounts of α-cubebene (4.4%), acoradiene, α-bulnesene, 4,10-dimethyl-7-isopropyl [4,4,0]-bicyclo-1,4-decadiene, β-chamigrene, thujopsene-(I2), and widdrol were also detected in the headspace of DtSTS9 cultures. In addition, *S*. *cerevisiae* strain expressing DtSTS10 gave rise to α-himachalene as the main product (83.59% of the peak area), together with small amounts of α-bisabolol (4.62%), β-sesquiphellandrene (2.9%), (+)-cuparene (2.0%), β-himachalene, cis-thujopsene, α-cedrene, acoradiene, γ-muurolene and δ-elemene. Interestingly, minor widdrol (2.14%) and bisabolol (4.62%) were detected in DtSTS10 culture based on GC-MS analysis, respectively. Yap et al., characterized the structures of (+)-torreyol, α-cardinol, and 1-napthalenol in genetically engineered *S*. *cerevisiae* cultures derived from *L*. *rhinocerotis* STSs [11]. The results suggested that the sesquiterpene alcohols might be generated from sesquiterpene precursors with the aid of endogenous cytochrome P450 monooxygenase of yeast [11]. 

## 4. Discussion

*Armillaria* spp. are devastating forest pathogens and they forage for hosts and achieve growing dispersal via root-like multicellular rhizomorphs [23]. Compared with other *Armillaria* species, natural melanized rhizomorphs of *D*. *tabescens* are rarely observed in laboratory or nature [25,26]. Herein, we sequenced and investigated the genome, performed RNA sequencing and directed discovery of sesquiterpene synthases in the strain CPCC 401429. The genome information would be helpful to investigate this unusual fungus. *Armillaria* species encode a large proportion of carbohydrate metabolism genes, and this could supply a strategy to bypass competition with other microbes of *Armillaria* spp. [23]. Compared to the genomes of other *Armillaria* fungi, the strain CPCC 401429 had a smaller genomic size (Table 2). The genomic features of *D*. *tabescens* had its own unique characteristics. There were 507 genes associated with carbohydrate metabolism, resulting in less genes associated with GHs, GTs, and CBMs among sequenced *Armillaria* strains. Slipos et al. (2017) reported that *Armillaria* species usually include large genomes for white rot saprotrophs evolved mostly by gene family diversification driven by transposable elements (TEs) proliferation [23,53,54]. Likewise, the *D*. *tabescens* exhibited relatively small proportion of TEs coding genes (2.89%), which is much lower than *Armillaria* species including *A*. *solidipes* (20.9%), *A*. *gallica* (32.6%), and *A*. *cepistipes* (34.8%) [23]. Meanwhile, the fungus CPCC 401429 encodes a wide array of genes involved in secondary metabolism of terpenoids and polyketides, which indicates that *D*. *tabescens* was a potential resource strain for polyketides and terpenoids production. Genome mining-based analysis of the sequenced *Armillaria* spp. and closely related species including *Guyanagaster necrorhizus* MCA 3950 and *Floccularia luteovirens* C10, demonstrated that melleolides biosynthetic gene cluster is commonly shared [4,55]. This indicated that these melleolide-coding BGCs is an interesting and evolutionary conserved feature of *Armillaria* genus or closely related species. This might provide crucial defense against predators or other microbes in adapting to a soil-borne lifestyle [23]. Conspicuously, the *mel* locus in *D*. *tabescens* is a shortest, delicate, and tightly clustered BGC responsible for melleolides biosynthesis [4]. Additionally, as to phylogenetic analysis of *D*. *tabescens* CPCC 401429, with the results of NR annotation, MLST-based phylogeny, and synteny analysis, we considered that *D*. *tabescens* was supposed to belong to the genus of *Desarmillaria* instead of *Armillaria* as it used to. It is noteworthy that the *D*. *tabescens* species formed a distinct group with *G*. *necrorhizus* MCA 3950, suggesting it would support the proposal of reattribution of *A*. *tabescens* and *A*. *ectypa* species in *Armillaria* subgenus *Desarmillaria* [25,26].

Basidiomycetous mushrooms are particularly adept at synthesizing a broad range of structurally diversified sesquiterpenes. Up to now, many Basidiomycetous sesquiterpene synthases have been characterized, exemplified as putative STSs from *O*. *olearius*, *C*. *puteana*, *L*. *rhinocerotus*, and *A*. *aegerita* [8,9,10,11]. With the recent revolution in genome-sequencing technology, a large number of Basidiomycetous genomes have been completed. Bioinformatic mining indicates that most genes/clusters of the phylum *Basidiomycota* are silent or cryptic under laboratory conditions [4,9,10]. The terpenoids and their synthases represent largely untapped “dark matter” that awaits to be characterized [1,2,4]. What are the ecological functions of STSs that play important roles in secondary metabolism during the lifecycle of mushrooms? What is the underlying mechanism that controls the functional evolution of STSs? Are these STSs unique to specific evolutionary lineage (taxa) or widely distributed? To answer these questions, functional characterization of increasing STSs encoded in the fungal genome is essential and indispensable.

As a cutting-edge discipline that drives original breakthroughs in the field of biopharmaceuticals, the rise of synthetic biology can promote drug leads discovery by designing new biosynthetic pathways. Therefore, mining and discovering of biosynthetic elements are urgently needed in synthetic biology technology [10]. Also, bioinformatic analysis and functional prediction of specific STSs would facilitate the screening of ideal candidate synthetic elements or bioprospecting for new sesquiterpenes. Since most Basidiomycetous mushrooms are difficult to manipulate in genetic transformation, *S*. *cerevisiae* yeast has emerged as a powerful host cell for production of fungal secondary metabolites and pathways reconstitution [11,56]. To our knowledge, this is the first report to document the endeavor of the yeast expression for mining the sesquiterpene repertoire in *Desarmillaria* species and *Armillaria* affinities. Phylogenetic analysis of characterized STSs reinforced that certain group of STSs shares highly conserved sequences, suggesting that they might catalyze the likely cyclization mechanism [9,10]. Thus, phylogenetic analysis might be useful to mine novel terpenoid synthases. In this study, noteworthy, half members STSs from the strain CPCC 401429 were distributed in minor group Clade IV. This group of STSs belongs to a new subfamily and only few STSs have been characterized. Our report on STSs derived from *D*. *tabescens* enriches the number of STSs of this subfamily and expands our knowledge of STSs diversity for *Desarmillaria* or close genus. Yeast cells expressing DtSTS9 and DtSTS10 yielded diverse sesquiterpene molecules, suggesting that this subfamily might be highly promiscuous producers. This highlights the potential of *Armillaria* in generating novel sesquiterpenoids. Accessing the basidiomycetous pool of STSs can be an immediate source of novel biocatalysts, or yield STSs that could be further specialized by directed evolution or combinatorial engineering. This also lays foundation for downstream in-depth deciphering catalytic mechanism of novel STSs.

## 5. Conclusions

In summary, we sequenced the genome of the fungus *D. tabescens* CPCC 401429 and identified twelve STSs encoding genes. Bioinformatic analysis and MLST-based phylogeny allowed us to support the reclassifying the fungus *D. tabescens* in *Desarmillaria* subgenus. Also, the large number of STS genes in the genome highlights the potential of the strain CPCC 401429 in generating varying sesquiterpenoids. In combination with transcriptomic analysis, two new STSs encoding genes were selected for yeast heterologous expression. Based on phylogenetic analysis and GC-MS identification, DtSTS9 and DtSTS10 belonging to the subfamily IV, are novel and very interesting sesquiterpene synthases with high potential for downstream research. This study also demonstrated that yeast based heterologous expression is an attractive approach for accessing the basidiomycete terpenoid chemistry.

## Figures and Tables

**Figure 1 jof-09-00481-f001:**
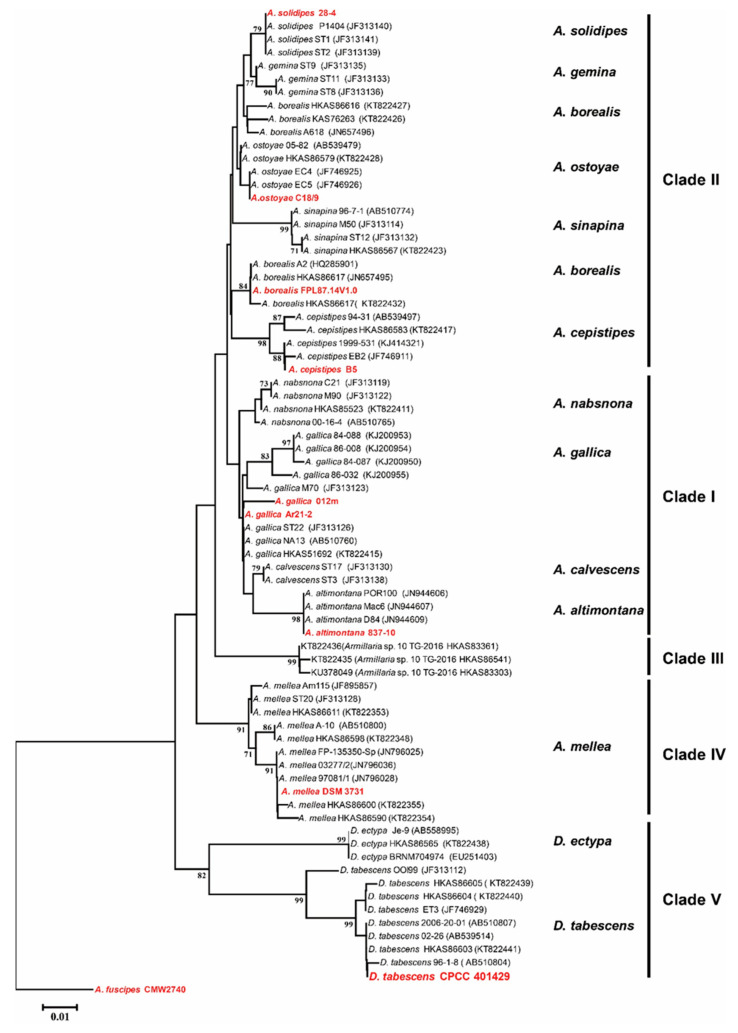
Dendrogram generated from elongation factor 1-α (Tef-1α) DNA sequence data showing the phylogenetic relationships of *Armillaria* species and lineages. The phylogenetic tree was constructed by the Neighbor-Joining method with MEGA 7.0 programme [47]. The bootstrap scores are based on 1000 reiterations. The Tef-1α DNA sequence of *A*. *fuscipes* CMW2740 acts as the outgroup.

**Figure 2 jof-09-00481-f002:**
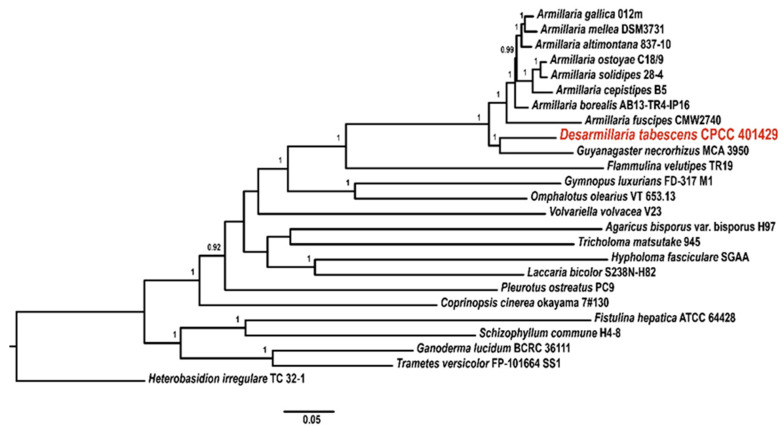
MLST-based phylogeny of the strain *D*. *tabescens* CPCC 401429 and twenty-four sequenced basidiomycetes encompassing white and brown rot fungi. The species of *Heterobasidion irregulare* TC 32-1 served as outgoup [23].

**Figure 3 jof-09-00481-f003:**
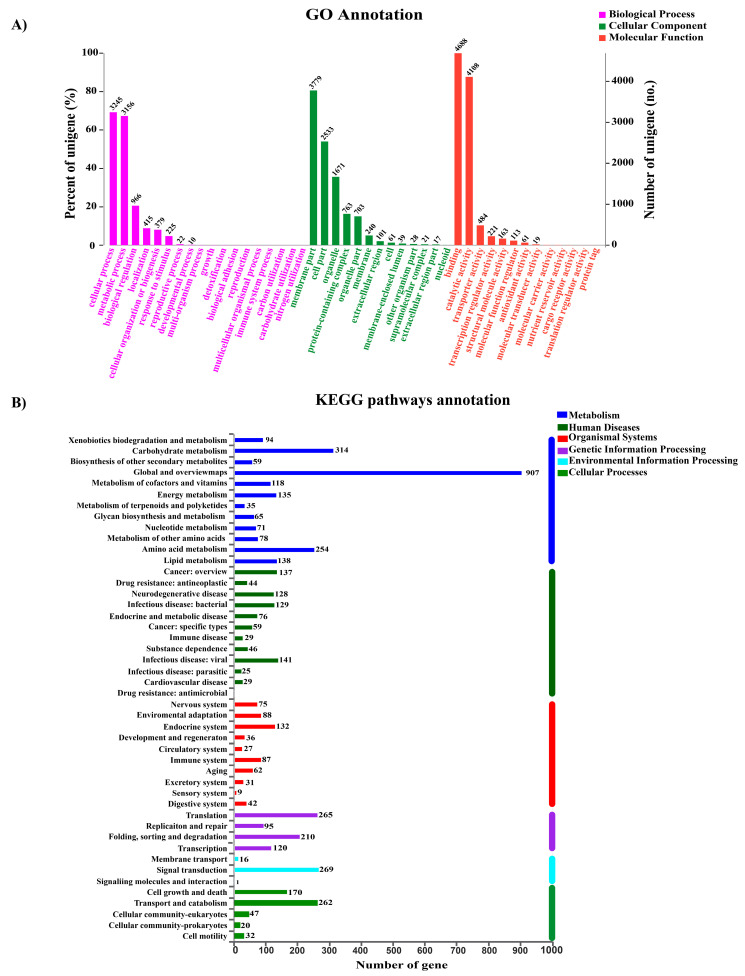
Gene function annotation of *D*. *tabescens* CPCC 401429 genome. (**A**) GO enrichment analysis of annotated genes from the strain CPCC 401429. (**B**) KEGG pathway annotation of the genome.

**Figure 4 jof-09-00481-f004:**
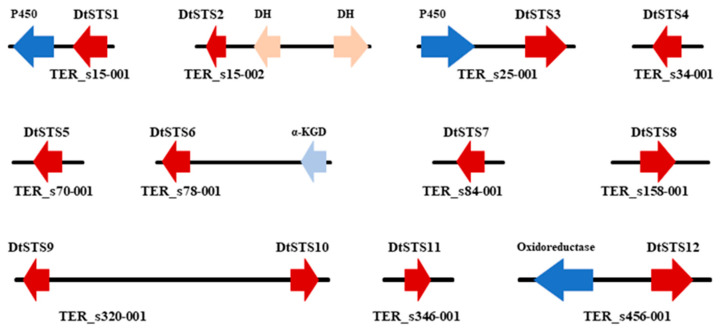
The genes (gene clusters) encoding sesquiterpene synthases (STSs) obtained from *D*. *tabescens* CPCC 401429. Note: sesquiterpene synthases DtSTS1-DtSTS1. DH, dehydrogenase; α-KGD, α-KG dependent dioxygenase; P450, P450 monooxygenase. The arrows represent the genes.

**Figure 5 jof-09-00481-f005:**
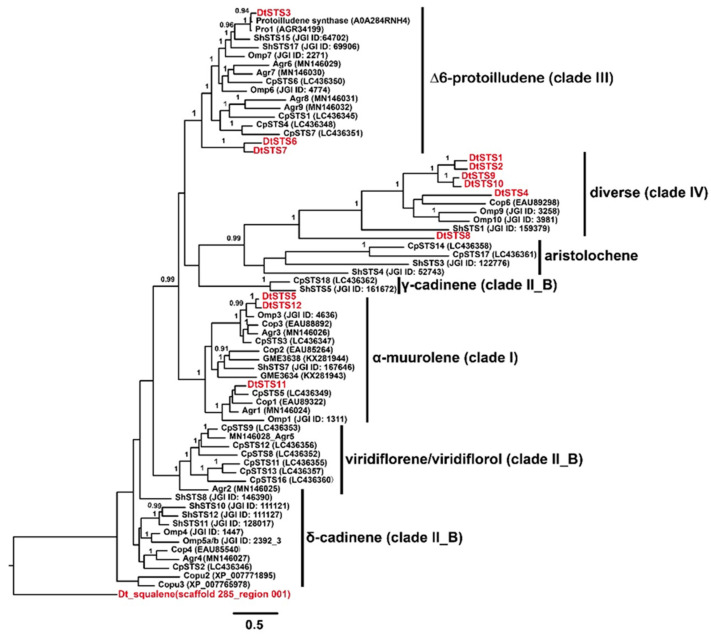
Phylogeny and distribution of sesquiterpene synthases obtained from *D*. *tabescens* CPCC 401429 with other basidiomycetous STSs. Rooted phylogenetic tree was constructed using MrBayes v3.2.6 [49]. The percentage of replicate trees in which the associated taxa clustered together in the bootstrap test (10,000,000 replicates) are shown next to the branches. The selected STSs were obtained from *C*. *pseudopinsitus* (CpSTS), *O*. *olearius* (Omp), *C*. *aegerita* (Agr), *C*. *puteana* (Copu), *C*. *cinerea* (Cop), *L*. *rhinocerotus* (GME), *S*. *hirsutum* (ShSTS), protoilludene synthases described from *A*. *ostoyae* (PRO1_ARMOS) and *A*. *gallica* (Pro1).

**Figure 6 jof-09-00481-f006:**
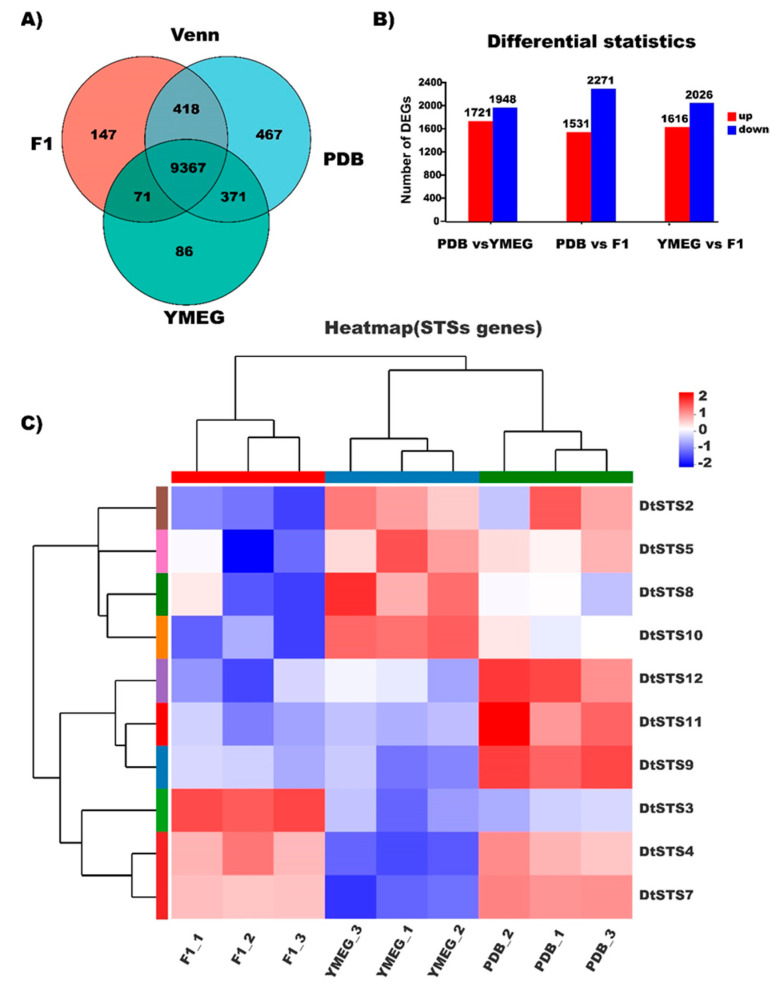
Mining transcriptome data for sesquiterpene synthases encoding genes in *D*. *tabescens* CPCC 401429. (**A**) Venn analysis of DEGs in three fermentation media. (**B**) Differentiated expression statistics of the transcriptome data of the strain CPCC 401429 in three fermentation media. (**C**) Heatmap of three samples including F1, YMEG, and PDB fermentation broth obtained by RNA−seq. F1−1, F1−2, and F1−3 are samples of F1 fermentation medium; YMEG−1, YMEG−2, and YMEG−3 are samples of YMEG fermentation medium; PDB−1, PDB−2, and PDB−3 are samples of PDB fermentation medium.

**Table 1 jof-09-00481-t001:** Strains, primers, genetically engineered yeast in the study.

Strains	Description	Source
*D*. *tabescens* CPCC 401429	Melleolides producing strain	[4]
*E. coli Trans* T1	*F-φ80 lac ZΔM15 Δ(lacZYA-arg F) U169 endA1 recA1 hsdR17(rk-,mk+) supE44λ- thi -1 gyrA96 relA1 phoA*	Transgen
*S. cerevisiae* BJ5464	*(MATα ura3-52 his3-Δ200 leu2- Δ1 trp1 pep4::HIS3 prb1 Δ1.6R can1 GAL*	[29]
BJ-CK	*S*. *cerevisiae* BJ 5464, carrying plasmid YET	This study
BJ-DtSTS9	*S*. *cerevisiae* BJ 5464, carrying plasmid YET-DtSTS9	This study
BJ-DtSTS10	*S*. *cerevisiae* BJ 5464, carrying plasmid YET-DtSTS10	This study
Plasmids	Description	Source
YET	tryptophan auxotrophic expressing plasmid of yeast	[29]
YET-DtSTS9	YET expressing plasmid with gene *DtSTS9*	This study
YET-DtSTS10	YET expressing plasmid with gene *DtSTS10*	This study
Primers	Sequence	Size (bp)
*DtSTS9*-F	atacaatcaactatcaactattaactatatcgtaataccatatgacactatctacagca	963 bp
*DtSTS9*-R	cttgataatggaaactataaatcgtgaaggcatgtttaaacttataatcccagctcgct	
*DtSTS10*-F	atacaatcaactatcaactattaactatatcgtaataccatatgactttatctacttcg	1005 bp
*DtSTS10*-R	cttgataatggaaactataaatcgtgaaggcatgtttaaacctatatgccaagctcgct	

**Table 2 jof-09-00481-t002:** Genome feature of the *Desarmillaria tabescens* CPCC 401429.

Genome Feature	No./Value
Genome size (Mb)	50.36
Number of scaffolds	713
Scaffold N_50_ (bp)	125,925
Scaffold N_90_ (bp)	33,591
Longest scaffold (bp)	839,203
Shortest scaffold (bp)	526
GC content (%)	47.14%
Protein-coding genes	15,145
Average gene length (bp)	1959
Complete BUSCOs (%)	92.4%
Genes of KEGG	3441
Genes of COG	9507
tRNA No.	255
rRNA No.	4
TE No.	438

**Table 3 jof-09-00481-t003:** Heterologous production of sesquiterpenes in yeast expressing STSs.

Rt (min)	Peak Area (%)	Name/Structure	Rt (min)	Peak Area (%)	Name/Structure
BJ-At_STS9	BJ-At_STS10
33.563	2.61	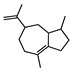 α-bulnesene	35.938	1.86	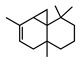 cis-thujopsene
34.167	1.29	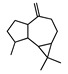 (+)-aromadendrene	38.618	0.54	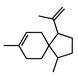 acoradiene
34.866	3.6	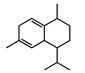 4,10-dimethyl-7-isopropyl [4,4,0]-bicyclo-1,4-decadiene	38.87	1.55	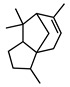 α-cedrene
35.971	11.25	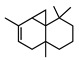 cis-thujopsene	39.39	0.88	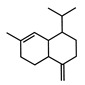 γ-muurolene
38.627	3.68	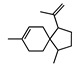 acoradiene	39.959	0.37	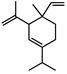 δ-elemene
39.699	4.4	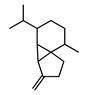 α-cubebene	41.567	83.59	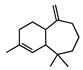 α-himachalene
40.243	1.66	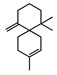 β-chamigrene	41.924	2.0	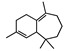 β-himachalene
41.511	0.68	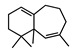 thujopsene-(I2)	43.573	2.9	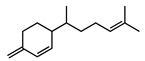 β-sesquiphellandrene
41.917	4.82	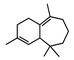 β-himachalene	45.864	2.0	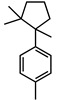 (+)-cuparene
45.865	1.41	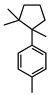 (+)-cuparene	54.483	4.62	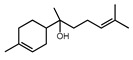 α-bisabolol
53.606	2.14	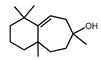 widdrol			

## Data Availability

All the raw sequence files are available in the National Center for Biotechnology Information under the OQ442799-OQ442810.

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
