# Peer review of "Genomic and Transcriptomic Approaches Provide a Predictive Framework for Sesquiterpenes Biosynthesis in Desarmillaria tabescens CPCC 401429"

_jof, 2023, doi:10.3390/jof9040481_

Round 1

Reviewer 1 Report

Dear Authors,

Here are my comments:

Title: Genomic and Transcriptomic Approaches Provide a Predictive Framework for Sesquiterpenes Biosynthesis in Desarmillaria tabescens CPCC 401429

Manuscript ID: jof-2255069

Comments

The paper describes the genome analysis of the fungi and the expression of terpene-coding genes. Authors should consider describing the details specifically as the flow of the manuscript is confusing. 

1. Abstract: This section needs major editing, explaining the problem statement and the hypothesis and aim of the study. 

2. Introduction: This section should start with the terpenes first, then describe their usage. Why the fungi is used in the study? The paragraphs in their current form look separated. 

3. Method: This section is well described and covers all the points. I would suggest authors modify the table 1. The authors cannot give the source as 'this study'. The study isn't published yet. Please give the proper source of each and every item used. Also, not all the items will go to one table. 

4. Results: Results are well described. 

5. Discussion: This section needs more information. Results are very promising, so it's important to discuss them in this section.

6. The last two paragraphs of the discussion look like a conclusion. Please edit. 

Reviewer 2 Report

Authors describes genomic and transcriptomic analysis  of basidiomycete mushroom D. tabescens and also functional characterization of 2 out of 12 sesquiterpene synthases encoded by the genome of this species.

The manuscript could be accepted for publication with some minor revisions.

Below are some points that need revision/clarifications:

1.line 54: "..nine draft genomes have been published to date" - currently there are several hundred

sequenced basidiomycota genomes

2. Not clear about annotation methods used. In the methods section  Augustus was mentioned but in Results they mention that 'protein-coding genes were predicted by MARKER2. What is MARKER2 - no citation we provided.

3. line 250: when estimating BUSCO completeness, what lineage was used  ('eukaryota', 'fungi' , 'basidiomycota') ?

4. line 254: Barrnap and rRNA-scan-SE should be swapped.

5. line 247:  N50 is probably 126Kb

6. line 403: what are the 'published reference spectra'  which authors used for terpene compound identifications ?

Reviewer 3 Report

Aim of this study is not measurable, please rewrite the aim in abstract after the background not at the end of the abstract. Also, add the conclusion of your research outcomes.
